# Expression and Polymorphisms of *SMAD1*, *SMAD2* and *SMAD3* Genes and Their Association with Litter Size in Tibetan Sheep (*Ovis aries*)

**DOI:** 10.3390/genes13122307

**Published:** 2022-12-07

**Authors:** Mingming Li, Na He, Ruizhe Sun, Yuting Deng, Xiaocheng Wen, Junxia Zhang

**Affiliations:** College of Agriculture and Animal Husbandry/Key Laboratory of Livestock and Poultry Genetics and Breeding on the Qinghai-Tibet Plateau, Ministry of Agriculture and Rural Affairs/Plateau Livestock Genetic Resources Protection and Innovative Utilization key Laboratory of Qinghai Province, Qinghai University, Xining 810016, China

**Keywords:** *SMAD1*, *SMAD2*, *SMAD3*, gene expression, polymorphism, litter size, Tibetan sheep

## Abstract

SMAD1, SMAD2, and SMAD3 are important transcription factors downstream of the TGF-β/SMAD signaling pathway that mediates several physiological processes. In the current study, we used cloning sequencing, RT-qPCR, bioinformatics methods and iMLDR technology to clone the coding region of Tibetan sheep genes, analyze the protein structure and detect the tissue expression characteristics of Tibetan sheep genes, and detect the polymorphisms of 433 Tibetan sheep and analyze their correlation with litter size. The results showed that the ORFs of the *SMAD1*, *SMAD2* and *SMAD3* genes were 1398 bp, 1404 bp and 1278 bp, respectively, and encoded 465, 467 and 425 amino acids, respectively. The SMAD1, SMAD2, and SMAD3 proteins were all unstable hydrophilic mixed proteins. *SMAD1*, *SMAD2* and *SMAD3* were widely expressed in Tibetan sheep tissues, and all were highly expressed in the uterus, spleen, ovary and lung tissues. Litter sizes of the genotype CC in the *SMAD1* gene g.10729C>T locus were significantly higher than that of CT (*p* < 0.05). In the *SMAD3* gene g.21447C>T locus, the genotype TT individuals showed a higher litter size than the CC and CT genotype individuals (*p* < 0.05). These results preliminarily demonstrated that *SMAD1*, *SMAD2* and *SMAD3* were the major candidate genes that affected litter size traits in Tibetan sheep and could be used as a molecular genetic marker for early auxiliary selection for improving reproductive traits during sheep breeding.

## 1. Introduction

Tibetan sheep (*Ovis aries*) are mainly distributed in the Qinghai–Tibet Plateau and are the main sources of income for local farmers and herdsmen. However, due to the long-term living in the low-oxygen environment of the plateau, Tibetan sheep have adapted very well to the harsh conditions. Moreover, the reproduction rate of Tibetan sheep is low, and the lambing ratio is only 105% [1]. The declining lambing rate is due to a low estrus rate and ovulation rate, but also is the result of the interaction between environmental and genetic factors [1]. Litter size is one of the most important economic traits in livestock. Because of their low heritability and the fact that the most quantitative traits are controlled by multiple genes, it is difficult to achieve rapid genetic improvement by traditional breeding methods alone [2]. An accurate performance assessment and genetic parameters estimation are the prerequisites for any successful genetic improvement program [3]. Studies have shown that the genetic improvement of livestock has been achieved by indirect selection of high-quality breeding stocks using marker-assisted selection (MAS) [4]. However, due to the influence of minor polygenes, only a small number of functional sites affecting specific traits have been identified. Therefore, it is particularly important to identify more functional genes and loci to enhance the understanding of the lambing performance of Tibetan sheep.

The TGF-β/SMADs signaling pathway is involved in early embryogenesis, bone formation, tissue repair, ovarian follicle development and follicle atresia and selection. This pathway also plays an important regulatory role in cell proliferation, differentiation, migration, apoptosis and immunity and endocrinology [5,6,7]. SMAD proteins are important transcription factors downstream of the TGF-β/SMADs signaling pathway, mediating intracellular transduction of extracellular TGF-β signals in cells [8,9]. SMAD proteins are divided into 3 categories according to their structural and functional characteristics: receptor-regulated Smads (R-Smads), including Smad1, 2, 3, 5, 8, 9; inhibitory Smads (I-Smads), including Smad6 and 7; and common mediator Smads (Co-Smad), including Smad4 [10,11]. *SMAD1*, *SMAD2* and *SMAD3* genes are widely distributed in different tissues and mediate multiple types of physiological processes [6,7,12]. *SMAD1* knockout mice exhibited embryonic death at E10.5 to E11.5, no or very little formation of primordial germ cells [13,14], decreased steroid hormone production, ovulation dysfunction and infertility [15]. By knocking out the *SMAD2* and *SMAD3* genes in female mice, mice developed follicular development and ovulation abnormalities, with significantly reduced fertility [16]. Female mice with *SMAD1/5* deletion showed endometrial defects leading to the development of cystic endometrial glands, endometrial epithelial hyperplasia during the implantation window and impaired apical–basal transformation, preventing embryo implantation and leading to infertility [17]. Tomic et al. [18] showed that *SMAD3*-deficient mice had significantly reduced fertility due to stunted follicle development, increased granulocyte apoptosis, and ovulation defects. The study of *SMAD1*, *SMAD2* and *SMAD3* genes has focused on mice and has also been involved in animals such as pigs [19] and cattle [20]. These genes have also been studied as candidate genes for fertility in sheep such as the small-tailed sheep [21,22] in China, the Hu sheep [23,24] in China, and the Garole × Malpura × Malpura [25,26] in India. In the current study, the structures of the genes were analyzed by cloning the *SMAD1*, *SMAD2* and *SMAD3* genes and using bioinformatics methods. RT-qPCR was used to detect the distribution and expression of *SMAD1*, *SMAD2* and *SMAD3* genes in different tissues of Tibetan sheep. The SNPs of *SMAD1*, *SMAD2* and *SMAD3* genes were genotyped by iMLDR technology, and the association between the polymorphisms and the litter size of Tibetan sheep were analyzed. The results of this study will provide a theoretical basis for the development of marker-assisted selection and breeding to improve the fertility of Tibetan sheep.

## 2. Materials and Methods

### 2.1. Experimental Animals and Sample Collection

In this experiment, 6 Tibetan ewes aged 6 months with a similar weight (35.32 ± 1.96 kg) and who were in good health were slaughtered at Xiangkameiduo Animal Husbandry Co., Ltd. in Gonghe county, Hainan Tibetan Autonomous Prefecture, Qinghai Province. The tissue samples including hypothalamus, pituitary, heart, liver, spleen, lungs, kidneys, ovaries, uterus, oviduct, rumen, duodenum and longissimus dorsi muscle were collected after slaughter. The samples were then quickly placed in liquid nitrogen, transported to the laboratory and stored in an ultra-low-temperature freezer at −80 °C. A total of 433 Tibetan ewes with a lambing record were selected from Xiangkameiduo Animal Husbandry Co., Ltd. From each Tibetan sheep, 5 mL of jugular vein blood was collected and placed in an anticoagulation vessel of heparin sodium, returned to the laboratory for low-temperature storage, and stored at −20 °C for backup.

### 2.2. RNA Isolation and cDNA Synthesis

Total RNA was isolated from 50 mg of tissue from the 13 samples (n = 3 for each tissue) using TRNzol Universal Reagent DP424 (TIANGEN Biotechnology, Co., Ltd., Beijing, China). The quality and integrity of the RNA was checked by agarose gel electrophoresis and the concentration of total RNA was measured by Nanodrop spectrophotometry (Gel DocTM XR+, Thermo Fisher Scientific Inc., Waltham, MA, USA). Further, cDNA was synthesized using total RNA (2 μg) by FastKing gDNA Dispelling RT SuperMix KR118 (TIANGEN Biotechnology, Co., Ltd., Beijing, China). After the termination of cDNA, each cDNA preparation was diluted four times with RNase-Free ddH_2_O and stored at −20 °C.

### 2.3. DNA Isolation

Genomic DNA was extracted from blood samples of 433 ewes using TIANamp Genomic DNA Kit DP348 (TIANGEN Biotechnology, Co., Ltd., Beijing, China) according to the instructions. The quantity and quality of DNA samples were measured with a NanoDrop2000 Nucleic Acid Protein Detector and by 1% agarose gel electrophoresis. Each DNA preparation was diluted to 10 ng/μL with RNase-Free ddH_2_O and stored at −20 °C.

### 2.4. Molecular Cloning of SMAD1, SMAD2 and SMAD3 cDNA

The primers of *SMAD1*, *SMAD2* and *SMAD3* were designed based on sequences of *SMAD1, SMAD2* and *SMAD3* from sheep by using the Primer Premier 5.0 software (Premier Biosoft International, Palo Alto, CA, USA) (Table 1). The primers were used for amplifying fragments of Tibetan sheep *SMAD1*, *SMAD2* and *SMAD3* cDNA. The polymerase chain reaction (PCR) amplification was performed in a total volume mixture of 25 μL, containing 12.5 μL 2 × Taq PCR Master Mix (TIANGEN Biotechnology, Co., Ltd., Beijing, China), 1 μL of cDNA template, 0.4 μL (10 μM) of forward primers, 0.4 μL (10 μM) of reverse primers and 10.7 μL of RNase-Free ddH_2_O. The reaction procedure involved predenaturation at 95 °C for 5 min, 30 cycles of denaturation at 95 °C for 30 s, annealing at *Tm* °C for 30 s and extension at 72 °C for 2 min, followed by a final extension at 72 °C for 5 min. The bands of desired sizes were excised and purified using a gel Extraction Kit (TIANGEN Biotechnology, Co., Ltd., Beijing, China) by following the instructions. The purified PCR products were then ligated with pMD19-T Vector (Takara, Japan), and transformed into competent Escherichia coli cells. Positive recombinant clones were identified by PCR and subsequently sent to a bio company (Shanghai Genesky Biotechnologies Inc., Shanghai, China) for sequencing.

### 2.5. Characteristics and Bioinformatics Analyses of SMAD1, SMAD2 and SMAD3 cDNA

The cDNA sequence acquired was spliced to obtain the core sequence of *SMAD1*, *SMAD2* and *SMAD3* cDNA by using DNAMAN software. Furthermore, the amino acid sequence was translated by using the DNAStar Lasergene 7.0 (Editseq). The physicochemical properties of the SMAD1, SMAD2 and SMAD3 proteins were predicted by using protparam (https://web.expasy.org/protparam/ accessed on 10 July 2022). The protein secondary structure of SMAD1, SMAD2 and SMAD3 in Tibetan sheep were predicted using SOPMA (NPS@: SOPMA secondary structure prediction (ibcp.fr)). To predict the protein structure of SMAD1, SMAD2 and SMAD3 in Tibetan sheep, comparative modeling was performed using Swiss Model (https://swissmodel.expasy.org/ accessed on 10 July 2022). The homology alignment of the amino acid sequences SMAD1, SMAD2 and SMAD3 in Tibetan sheep was compared with amino acids from other species in the GenBank database using the DNAStar Lasergene 7.0 (MegAlign). A phylogenetic tree was constructed by the neighbor-joining method by using the MEGA 6.0 software (Borland company, Scotts Valley, CA, USA).

### 2.6. Real-Time Quantitative PCR

For RT-qPCR of the *SMAD1*, *SMAD2* and *SMAD3* genes, the primers used for *SMAD1*, *SMAD2*, *SMAD3* and *GAPDH* expression were designed using primer premier 5.0 software (Premier Biosoft International, Palo Alto, CA, USA) based on the sheep sequences *SMAD1*, *SMAD2*, *SMAD3* and *GAPDH* obtained from the NCBI database (Table 1). *GAPDH* was used as a housekeeping gene. The expressions of *SMAD1*, *SMAD2*, *SMAD3* genes in 13 different tissues of Tibetan sheep were quantified by RT-qPCR. The RT-qPCR was performed on Bio-Rad CFX96 Real-Time PCR System (Bio-Rad Laboratories, Hercules, CA, USA) using 96-well plates. Each 20 µL real-time qPCR reaction system contained 2 µL of cDNA, 10 µL 2 × SuperReal Color PreMix (TIANGEN Biotechnology, Co., Ltd., Beijing, China), 0.5 μL 40 × Dilution Buffer, 0.6 µL of each forward and reverse primer and 6.3 µL of ddH_2_O. The RT-qPCR protocol was as follows: 95 °C for 15 min followed by 40 cycles of 95 °C for 10 s and 60 °C for 32 s, melting the amplification with constant heating from 65 °C for 5 s to 95 °C to obtain the melting curve. All samples were assayed in triplicate. The obtained data were normalized by *GAPDH* and calculated using the 2^−∆∆Ct^ method.

### 2.7. Genotyping

Genomic DNA was extracted from the venous jugular blood. The selected 10 SNPs in *SMAD1*, *SMAD2* and *SMAD3* were genotyped with the method of polymerase chain reaction (PCR) ligase detection reaction (LDR) on an ABI 3730XL Sequence Detection System (Applied Bio-systems, Waltham, MA, USA), with technical support from the Shanghai Genesky Biotechnologies Inc. The primer sequences used for the PCR reaction are described in Table 2. The PCR reaction was carried out in 20 μL of 1 × GC-I buffer (Takara), 3.0 mM Mg^2+^, 0.3 mM dNTP, 1U HotStarTaq polymerase (Qiagen Inc., Germantown, MD, USA), 1 μL of sample DNA and 1 µL of each primer. PCR amplification was performed as follows: 95 °C for 2 min and 11 cycles at 94 °C for 20 s, 65 °C for 40 s, 72 °C for 1.5 min, and 24 cycles at 94 °C for 20 s, 59 °C for 30 s, and 72 °C for 1.5 min and a final extension at 72 °C for 2 min. PCR products were purified with 5 U of Shrimp Alkaline Phosphatase and 2 U of Exonuclease I in a 37 °C warm bath for 1 h and then inactivated at 75 °C for 15 min to degrade excess dNTPs and primers.

Two allele-specific probes and one fluorescently-labeled probe were used for LDR (Table 2). The reader is referred to Yuan [27] for detailed principles on the typing step utilized. LDR was carried out in 1 μL of 10 × binding buffer, 0.25 μL of thermostable Taq DNA ligase, 0.4 μL of 1 mM 5′ ligation primers mixture, 0.4 μL of 2 mM 3′ ligation primers mixture, 5 μL of multiplex PCR product, and 3 μL of ddH_2_O. LDR was performed as follows: 38 cycles of 94 °C for 1 min and 56 °C for 4 min, followed by storage at 4 °C. Half a microliter of the reaction mixtures was denatured at 95 °C for 5 min in 9 μL Hi-Di formamide along with 0.5 μL of the LIZ-500 size standard and run on the ABI 3730XL genetic analyzer. Data analysis was achieved using GeneMapper Software v4.0 (Applied Biosystems, New York, NY, USA).

### 2.8. Statistical Analysis

A one-way analysis of variance (ANOVA) was used to detect whether there were statistical differences in the average mRNA expression of *SMAD1*, *SMAD2* and *SMAD3* genes in 13 tissues of Tibetan sheep. Genotype frequencies, allelic frequencies, polymorphism information content (*PIC*), effective allele number (*Ne*), gene homozygosity (*Ho*), gene heterozygosity (*He*) and Hardy–Weinberg equilibrium (*HWE*) were directly calculated with reference to Zhao et al. [28]. The association between *SMAD1*, *SAMD2* and *SMAD3* genotypes and the litter size of Tibetan ewes was analyzed according to a general linear model (GLM) program. Based on the characteristics of the sheep, the statistical model was as follows:*Y*_*ijn*_ = *μ* + *P*_*i*_ + *G*_*j*_ + *I*_*PG*_ + *e*_*ijn*_
where *Y_ijn_* is the phenotypic observation value; *μ* is the overall population mean; *P_i_* is the fixed effect of the *i*th parity (*i* = 1, 2 or 3); *G_j_* is the effect of the *j*th genotype (*j* = 1, 2 or 3); *I_PG_* is the interactive effect of parity and genotype and e *ijn* represents random error; assuming that *e_ijkl_* is independent of each other, obeying the N (0, σ2) distribution. Moreover, the linkage disequilibrium (LD) of identified SNPs was performed using Haploview software [29]. Results with a *p* < 0.05 were considered significantly different. Results are presented as means ± standard error of means (SEM). All statistical analyses were performed by using Statistical Package for Social Science (SPSS) version 23.0 for Windows (SPSS, IBM, Armonk, NY, USA).

## 3. Results

### 3.1. Molecular Cloning and Sequence Analysis of SMAD1, SMAD2 and SMAD3

The Tibetan sheep *SMAD1*, *SMAD2* and *SMAD3* genes were successfully cloned in the experiment, and the sequence analysis results showed that the ORFs of *SMAD1*, *SMAD2* and *SMAD3* were 1398 bp, 1404 bp and 1278 bp, respectively, and encoded 465, 467 and 425 amino acids, respectively. The molecular weight of the *SMAD1*, *SMAD2* and *SMAD3* genes were 52.24 kD, 52.31 kD and 48.08 kD, respectively, and the isoelectric points were 6.90, 6.13 and 6.73, respectively. The total number of negatively-charged residues (Asp + Glu) of the SMAD1, SMAD2 and SMAD3 proteins were 44, 44 and 43, respectively, while the total number of positively-charged residues (Arg + Lys) were 42, 42, and 41, respectively. The calculated lipid index was 65.35, 74.07, and 74.52, respectively. The grand average of hydropathicity was −0.568, −0.444, and −0.447, respectively, indicating that they were all hydrophilic proteins. The instability index (II) was 60.84, 53.30, and 53.26, respectively, indicating that they were all unstable proteins.

### 3.2. Multiple Sequences Alignment and Phylogenetic Analysis

The SMAD1, SMAD2 and SMAD3 polypeptide sequences of Tibetan sheep were compared with those of other animals by MegAlign software (Figure 1). The Tibetan sheep SMAD1 polypeptide sequence displayed a high percentage of identity with other species such as *Ovis aries* (100.0%), *Bos mutus* (100.0%), *Bos taurus* (99.8%), *Homo sapiens* (99.8%), *Sus scrofa* (99.8%), *Macaca mulatta* (99.8%), *Canis lupus familiaries* (99.6%), *Pan troglodytes* (99.4%), *Mus musculus* (98.9%), *Gallus gallus* (97.2%) and *Maylandia zebra* (90.9%) (Figure 1A). The percentage of the peptide sequences homology of Tibetan sheep SMAD2 displayed a high proportion of identity with *Ovis aries* (100.0%), *Homo sapiens* (100.0%), *Sus scrofa* (100.0%), *Canis lupus familiaries* (100.0%), *Pan troglodytes* (100.0%), *Macaca mulatta* (100.0%), *Bos mutus* (99.8%), *Mus musculus* (99.6%), *Bos taurus* (99.6%), *Gallus gallus* (99.4%), *Maylandia zebra* (90.8%) and a few more with over 90% similarity (Figure 1B). The Tibetan sheep SMAD3 was most similar to *Ovis aries* (100.0%), *Homo sapiens* (100.0%), *Sus scrofa* (100.0%), *Mus musculus* (100.0%), *Canis lupus familiaries* (100.0%), *Pan troglodytes* (100.0%), *Bos taurus* (99.8%), *Macaca mulatta* (99.8%), *Bos mutus* (99.8%), *Gallus gallus* (99.1%) and *Maylandia zebra* (96.5%) (Figure 1C).

The phylogenetic analyses of the *SMAD1*, *SMAD2* and *SMAD3* nucleotide sequences were performed using MEGA 6.0 through the neighbor-joining method (Figure 2). In the phylogenetic tree of *SMAD1* (Figure 2A), *SMAD2* (Figure 2B) and *SMAD3* (Figure 2C) genes, the sequence identity at the nucleotide level of Tibetan sheep (red text) showed a closer similarity with *Ovis aries* than with other ruminants such as *Bos taurus* and *Bos mutus*. Furthermore, the sequence of the Tibetan sheep showed a divergence with *Homo sapiens*, *Sus scrofa*, *Mus musculus*, *Canis lupus familiaries*, *Pan troglodytes*, *Macaca mulatta*, *Gallus gallus* and *Maylandia zebra*.

### 3.3. Protein Structure Prediction of SMAD1, SMAD2 and SMAD3

The secondary structures of SMAD1, SMAD2 and SMAD3 proteins were predicted using the online software SOPMA (Figure 3), and the secondary structures of SMAD1 (Figure 3A) proteins included 252 random coils, 105 α helixes, 83 extended strands and 25 β turns, accounting for 54.19%, 22.58%, 17.85% and 5.38%, respectively. The secondary structure of SMAD2 (Figure 3B) proteins included 269 random coils, 95 α helixes, 83 extended strands and 20 β turns, accounting for 57.60%, 20.34%, 17.77% and 4.28%, respectively. The secondary structure of SMAD3 (Figure 3C) proteins included 232 random coils, 100 α helixes, 77 extended strands and 16 β turns, accounting for 54.59%, 23.53%, 18.12% and 3.76%, respectively. It could be inferred that the four structures of the random coil, α helix, extended strand and β turn were the main bodies that made up the secondary structure of the SMAD1, SMAD2 and SMAD3 proteins. The tertiary structural models of the SMAD1 (Figure 4A), SMAD2 (Figure 4B) and SMAD3 (Figure 4C) proteins were established by the SWISS-MODEL software, and the results showed that they were consistent with the predicted secondary structures.

### 3.4. The mRNA Expression of SMAD1, SMAD2 and SMAD3 Genes in the Different Tissues of Tibetian Sheep

The RT-qPCR was used to investigate the tissue distributions of the *SMAD1*, *SMAD2* and *SMAD3* genes in Tibetan sheep. The results showed that the mRNA of the *SMAD1*, *SMAD2* and *SMAD3* genes exhibited a widespread expression in all thirteen tissues including the hypothalamus, pituitary, heart, liver, spleen, lung, kidney, ovary, uterus, oviduct, rumen, duodenum and longissimus dorsi (Figure 5).

The expression of the *SMAD1* gene in the spleen was significantly higher than in other tissues (*p* < 0.05) and its expression was significantly higher in the uterus than in the hypothalamus, ovary, oviduct, duodenum, rumen, pituitary, heart, liver, kidney and longissimus dorsi (*p* < 0.05). *SMAD1* gene expression in the lung was significantly higher than in the oviduct, duodenum, rumen, pituitary, heart, liver, kidney, heart and longissimus dorsi (*p* < 0.05). *SMAD1* gene expression in the oviduct was significantly higher than in the kidney, heart and longissimus dorsi (*p* < 0.05). However there were not significant differences among the lung, hypothalamus and ovary (*p >* 0.05). The mRNA expression of the *SMAD2* gene was significantly higher in the lung than in other tissues (*p* < 0.05), and *SMAD2* gene expression in the spleen was significantly higher than in the uterus, duodenum, liver, kidney, oviduct, rumen, pituitary, heart, hypothalamus and longissimus dorsi (*p* < 0.05). However, there were not significant differences among the ovary and uterus (*p >* 0.05).

The mRNA expression of the *SMAD3* gene in the uterus was significantly higher than in other tissues (*p* < 0.05). *SMAD3* gene expression in the spleen was significantly higher than in the ovary, lung, oviduct, duodenum, pituitary, kidney, rumen, hypothalamus, liver, heart and longissimus dorsi muscle (*p* < 0.05). *SMAD3* gene expression in the ovary was significantly higher than that in the lung, oviduct, duodenum, pituitary, kidney, rumen, hypothalamus, liver, heart and longissimus dorsi muscle (*p* < 0.05). *SMAD3* gene expression in the lungs was significantly higher than in the oviduct, duodenum, pituitary, kidney, rumen, hypothalamus, liver, heart and longissimus dorsi muscle (*p* < 0.05), and the expression in the oviduct was significantly higher than in the pituitary, kidney, rumen, hypothalamus, liver, heart and longissimus dorsi muscle (*p* < 0.05).

### 3.5. Population Genetic Analysis of Polymorphisms in Tibetan Sheep SMAD1, SMAD2 and SMAD3

Four SNPs of the *SMAD1* gene were screened: g.45975G>A and g.45823T>C were located in the 2nd exon region; g.10729C>T was located in the 5th exon; and g.440G>A was located in the 7th exon, all of which were synonical mutations (Table 3). Three genotypes were detected in g.45975G>A including GG, GA and AA, of which the dominant genotypes were GG and the dominant alleles were G (Table 3). The homozygosity (*Ho*), heterozygosity (*He*) and effective allele numbers (*Ne*) for g.45975G>A were 0.75, 0.25 and 1.34, respectively (Table 4). Only two genotypes were detected at the g.440G>A, namely GG and GA, with the dominant genotype being GG and the dominant allele being G (Table 3). The *Ho*, *He* and *Ne* were 0.92, 0.08, and 1.08, respectively (Table 4). TT, TC and CC genotypes were detected at the g.45823T>C mutation site, where the dominant genotype was CC and the dominant allele was C (Table 3). The *Ho*, *He*, and *Ne* were 0.63, 0.37 and 1.58, respectively (Table 4). Only CC and CT were detected at the g.10729C>T mutation site, with the dominant genotype being CC and the dominant allele being C (Table 3). The *Ho*, *He* and *Ne* were 0.96, 0.04 and 1.04, respectively (Table 4). *SMAD2* gene screening obtained a synonical mutation site located in exon 8, and three genotypes were detected at the g.14946G>A mutation site, namely, GG, GA and AA. The dominant genotype was GG and the dominant allele was G (Table 3). The *Ho*, *He* and *Ne* were 0.58, 0.42 and 1.73, respectively (Table 4). *SMAD3* gene screening yielded a total of 5 SNPs, of which g.5133C>T was located in the 4th exon region, g.18965C>T and g.18905T>C were located in the 6th exon region and g.21447C>T and g.21551A>G were located in the 7th exon region. Except for g.18905T>C, which was a missense mutation, the rest of the SNPs were synonical mutations. Three genotypes were detected at the three mutant loci of g.21447C>T, g.18965C>T and g.5133C>T, namely, CC, CT, and TT. The dominant genotype of the first two was CC and the dominant allele was C, while the dominant genotype at the third mutant site was CT and dominant allele was C (Table 3). On these three SNPs, the *Ho* was 0.69, 0.75 and 0.54, the *He* was 0.31, 0.25 and 0.46 and the *Ne* was 1.45, 1.34 and 1.85, respectively (Table 4). At the g.18905T>C mutation site, three genotypes of TT, TC and CC were detected, with the dominant genotype being CC and the dominant allele being C (Table 3). At this mutation site, the *Ho*, *He*, and *Ne* were 0.56, 0.44, and 1.79, respectively (Table 4). At the g.21551A>G mutation site, AA, AG and GG were detected, with the dominant genotype being GG and the dominant allele being G (Table 3). At this mutation site, the *Ho*, *He*, and *Ne* were 0.56, 0.44, and 1.79, respectively (Table 4). The results of the *PIC* analysis showed that the *SMAD1* gene g.45823T>C locus, the *SMAD2* gene g.14946G>A locus and the *SMAD3* gene g.21447C>T, g.5133C>T, g.18905T>C and g.21551A>G loci were all moderate polymorphic (0.25 < *PIC* < 0.5). The *SMAD1* gene g.45975G>A, g.440G>A and g.10729C>T sites and the *SMAD3* gene g.18965C>T sites were all in low-degree polymorphism (*PIC* < 0.25) (Table 4). The χ^2^ test showed that the genotype distribution of all the mutation loci of all genes were in a Hardy–Weinberg equilibrium state (*p* > 0.05) (Table 4).

### 3.6. Association Analysis of SMAD1, SMAD2 and SMAD3 Genes between SNPs and Litter Size in Tibetian Sheep

The association analysis of the genotypes with litter size was conducted in this study (Table 5). For g.10729C>T in the *SMAD1* gene, individuals with genotype CC had a significantly higher litter size compared with genotype CT (*p* < 0.05). For g.45975G>A, g.45823T>C and g.440G>A in *SMAD1*, there was no significant difference between disparate genotypes in relation to the litter size (*p >* 0.05). The g.14946G>A locus genotype in the *SMAD2* gene was not significantly correlated with the litter size in Tibetan sheep. Ewes carrying genotype TT of the g.21447C>T locus in the *SMAD3* gene had a significantly higher litter size than those carrying genotype CT and CC (*p* < 0.05). For g.5133C>T, g.18965C>T, g.18905T>C and g.21551A>G in the *SMAD3* gene, there was no significant difference between disparate genotypes in relation to the litter sizes (*p >* 0.05).

### 3.7. Linkage Disequilibrium and Haplotype Analysis of SNPs in SMAD1 and SMAD3 Genes in Tibetan Sheep

The results of the linkage disequilibrium analysis of SNPs loci of the *SMAD1* and *SMAD3* gene are shown in Figure 6. Two blocks were found in the *SMAD1* gene (Figure 6A). The first block with g.440G>A and g.10729C>T variants of the *SMAD1* gene was completely linked; three haplotypes (AC, GC and GT) were found in the Tibetan sheep population, and the frequency of the GC haplotype was the highest (0.94), whereas that of the GT haplotype was the lowest (0.02) (Table 6). The second block with g.45823T>C and g.45975G>A variants of the *SMAD1* gene was completely linked; three haplotypes (CA, CG and TG) were found in the Tibetan sheep population, and the frequency of the CG haplotype was the highest (0.71), whereas that of the CA haplotype was the lowest (0.16) (Table 6). Seven blocks were found in the *SMAD3* gene (Figure 6B). The first block with g.5133C>T and g.18905T>C variants of the *SMAD3* gene was completely linked; three haplotypes (CC, CT and TC) were found in the Tibetan sheep population, and the frequency of the TC haplotype was the highest (0.61) (Table 6). The other six blocks between g.18905T>C, g.18965C>T, g.21447C>T and g.21551A>G variants of the *SMAD3* gene were completely linked; four haplotypes (CCCG, CCTG, CTCG and TCCA) were found in the Tibetan sheep population, and the frequency of the CCCG haplotype was the highest (0.59), whereas that of the CTCG haplotype was the lowest (0.20) (Table 6).

## 4. Discussion

SMADs genes are involved in important physiological processes such as disease, immune regulation, growth and development, wound healing, cartilage and bone development and maintenance through the TGFβ superfamily. These genes also regulate the proliferation, differentiation, maturation, adhesion, atresia, apoptosis and steroid hormone production of germ cells [6,7,30]. Therefore, understanding the structure and function of SMADs genes is of great significance for the study of animal growth, development and reproduction, and can avoid the differences caused by artificial selection on gene evolution. In this study, the *SMAD1*, *SMAD2* and *SMAD3* genes of Tibetan sheep were successfully cloned. The ORFs were found to contain 1398 bp, 1404 bp and 1278 bp, respectively, and encoded 465, 467 and 425 amino acids, respectively. The detection of ORFs is an important step in finding protein-coding genes in genomic sequences [31]. The amino acid sequences of SMAD1, SMAD2 and SMAD3 in Tibetan sheep were more than 90% homologous to the 11 selected species and 100% similar to sheep. Compared with non-ruminants, the *SMAD1*, *SMAD2*, and *SMAD3* genes are more conserved in ruminants. The average hydrophilic indices of the SMAD1, SMAD2 and SMAD3 proteins were all negative, and the instability coefficients were all >40, all of which were unstable hydrophilic proteins. The SMAD1, SMAD2 and SMAD3 proteins were all composed of a random coil, α helix, extended strand and β turns, all of which were mixed proteins. Thus, the SMAD1, SMAD2, and SMAD3 proteins were all unstable hydrophilic mixed proteins.

SMAD1, as an important transcription factor downstream of the TGF-β/SMADs signaling pathway, is widely distributed in different tissues and mediates several types of physiological processes [12]. Tian et al. [22] found that the *SMAD1* gene was expressed in the whole-body tissues of small-tailed Han sheep with different fecundity, and the expression of ovaries in the single-lamb population of small-tailed Han sheep was significantly higher than that of the multi-lamb population (*p* < 0.01). The expression of the hypothalamus and pituitary in the small-tailed Han sheep population was also significantly higher than that of the multi-lamb population (*p* < 0.05). Niu et al. [32] used RT-PCR technology to detect the expression profile of *SMAD1* mRNA in yak tissue and found that the *SMAD1* gene was broadly expressed. Ao et al. [33] found that the *SMAD1* gene was expressed in various tissues of Qianbei Ma Goats. The above is consistent with the use of RT-qPCR to detect extensive expression in various tissues of Tibetan sheep in this study. The relative mRNA expression of the *SMAD1* gene in the ovary and uterus differed from that found by Tian et al. [22], possibly due to different varieties. SMAD2/SMAD3 is a key molecule in the classic TGF-β/Smads signaling pathway, which can directly regulate the growth and development of follicles and ovulation by regulating the maturation of oocytes, the proliferation, differentiation and apoptosis of granulocytes, the regulation of human FSHβ promoter activity, the maintenance of normal aromatase expression levels, and the maintenance of the normal secretory function of the ovaries [34,35]. Researchers have found in cultured ovarian cells that activating the SMAD2/3 signal transduction pathway induces cell expansion and protects amplified ovulus cells from apoptosis, while also inducing the expression of expansion-related genes [36]. Zheng [37] detected the results of the *SMAD2* gene expression profiling in various tissues of Hu sheep, which was consistent with our results. The relative expression of *SMAD1*, *SMAD2* and *SMAD3* genes in Tibetan sheep lung may be related to the hypoxic fitness of Tibetan sheep.

Single nucleotide polymorphisms (SNPs) are nucleotide changes at individual genomic loci between important subgroups of populations and are the main source of genomic variation. As molecular markers, SNPs are becoming increasingly attractive because they are associated with many important productive traits in livestock [38]. Missense mutations lead to defective protein translation and are associated with many diseases [39]. Synonymous mutations do not usually cause changes in the function of the coding protein, but some synonymous mutations have been shown to affect the secondary structure of the mRNA, the folding and conformation of proteins, etc., [40], which in turn leads to changes in gene function and phenotype. Recent studies have shown that synonymous mutations play a role in the regulation of sheep production traits [41,42,43]. In the present study, through the association analysis of a total of 10 SNPs genotypes of the *SMAD1*, *SMAD2* and *SMAD3* genes and litter size in Tibetan sheep population, it was found that the two synonymous mutation sites of the *SMAD1* gene g.10729C>T and the *SMAD3* gene g.21447C>T were significantly associated with litter size, while the *SMAD3* gene g.18905T>C was not significantly associated with the litter size. A population genetic analysis found that the *SMAD1* gene g.45823T>C locus, *SMAD2* gene g.14946G>A locus and *SMAD3* gene g.21447C>T, g.5133C>T, g.18905T>C and g.21551A>G loci were all moderate polymorphisms (0.25 < *PIC* < 0.5). The 10 SNPs loci were all in the Hardy–Weinberg equilibrium state in the Tibetan sheep population, indicating that the different genotypes of the 10 SNPs were widely present in the Tibetan sheep and could be stably passed on to future generations. Xu et al. [23] have found that the rs40635766 mutation located in the intron region of the *SMAD1* gene is closely related to the fertility of Hu sheep and may be a dominant gene that controls the multi-lamb reproduction of Hu sheep. Tian et al. [22] found that the litter size in the first three parities of the g.12487190G>T locus TT genotype small-tailed Han sheep ewes located in the intron 5 region was significantly higher than that of the GT and GG genotype ewes (*p* < 0.05). In this study, for g.10729C>T in the *SMAD1* gene, individuals with genotype CC had a significantly larger litter size compared with genotype CT (*p* < 0.05). Ewes carrying genotype TT of the g.21447C>T locus in the *SMAD3* gene had a significantly larger litter size than those carrying genotype CT and CC (*p* < 0.05). Therefore, the *SMAD1* and *SMAD3* gene could be used as molecular markers for improving litter size in Tibetan sheep. Linkage disequilibrium analysis can detect the interplay between multiple genetic loci, and haplotypes include the simple addition and interaction of multiple SNPs to more effectively interpret the genetic information of phenotypic traits [44]. A total of six kinds of haplotypes in *SMAD1* and seven in *SMAD3* were obtained in this study, respectively. However, further studies on the mechanism of *SMADs* affecting the litter size of Tibetan sheep are required.

## 5. Conclusions

In this study, we found that the ORFs of the *SMAD1*, *SMAD2* and *SMAD3* genes were 1398 bp, 1404 bp and 1278 bp, respectively, and encoded 465, 467 and 425 amino acids, respectively. The SMAD1, SMAD2, and SMAD3 proteins were all unstable hydrophilic mixed proteins and had the closest relationship to the sheep. The *SMAD1*, *SMAD2* and *SMAD3* genes were widely expressed in Tibetan sheep tissues, and all were highly expressed in the uterus, spleen, ovary and lung tissues. The litter size of the genotype CC in the *SMAD1* gene g.10729C>T locus was significantly higher than that of CT (*p* < 0.05), and in the *SMAD3* gene g.21447C>T locus, the genotype TT individuals showed higher litter sizes than the CC and CT genotype individuals (*p* < 0.05). Therefore, the *SMAD1* and *SMAD3* genes may be important candidate genes affecting litter size in Tibetan sheep, and the *SMAD1* gene g.10729C>T locus and the *SMAD3* gene g.21447C>T locus have important values for molecular marking assistant selection of litter sizes in Tibetan sheep.

## Figures and Tables

**Figure 1 genes-13-02307-f001:**
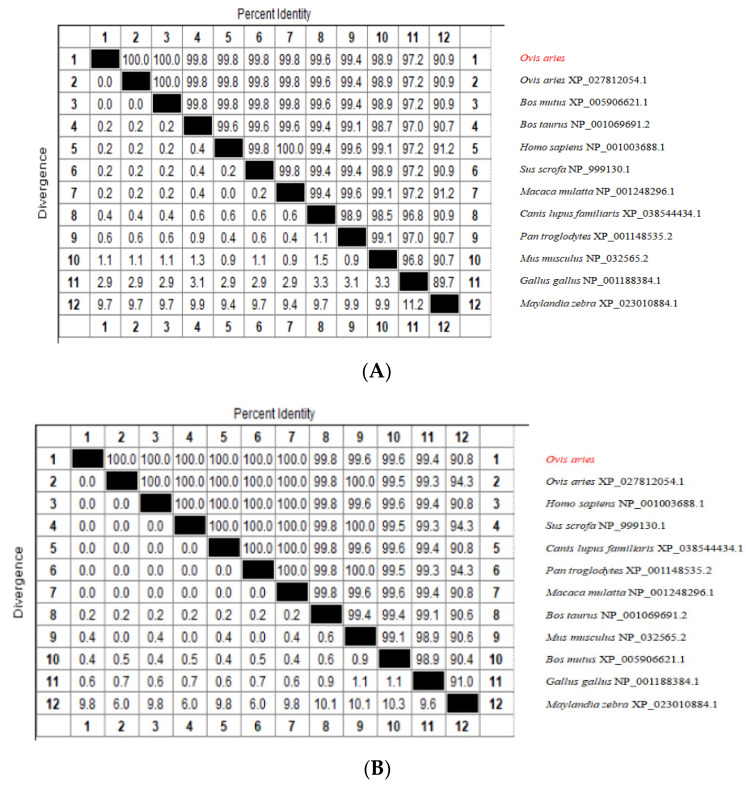
SMAD1 (**A**), SMAD2 (**B**) and SMAD3 (**C**) polypeptide sequence identity of Tibetan sheep with 11 other species. The red highlight text is the species studied in this article.

**Figure 2 genes-13-02307-f002:**
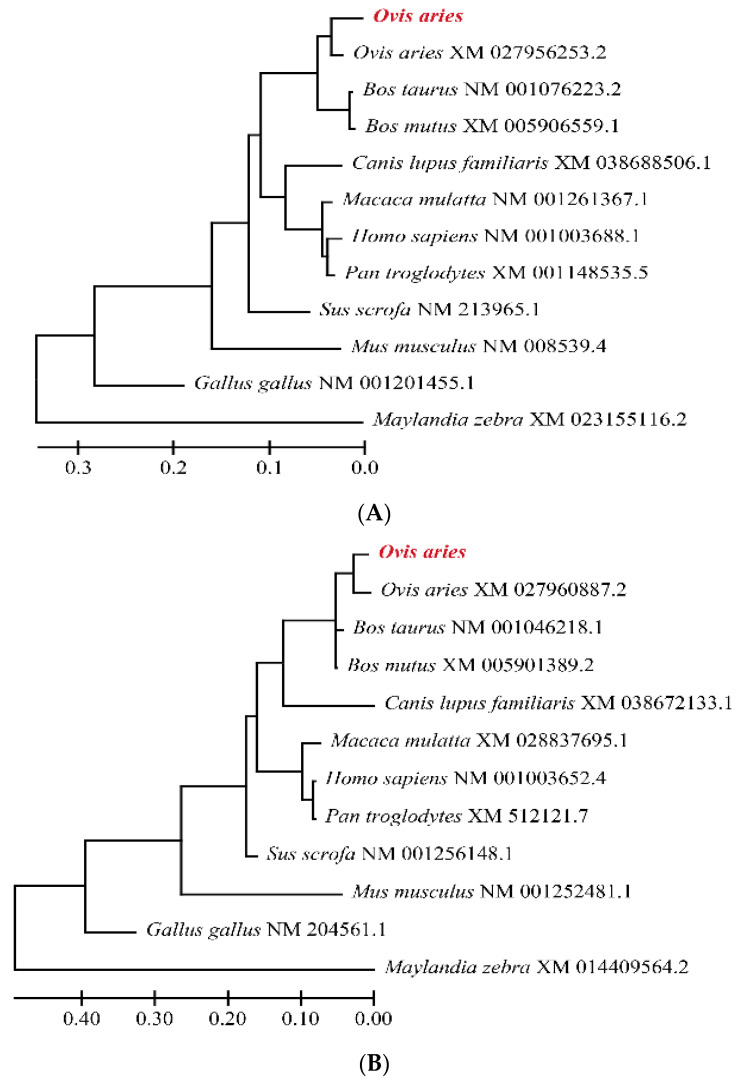
Phylogenetic tree of *SMAD1* (**A**), *SMAD2* (**B**) and *SMAD3* (**C**) of Tibetan sheep with 11 other species. The red highlighted text is the species studied in this article.

**Figure 3 genes-13-02307-f003:**
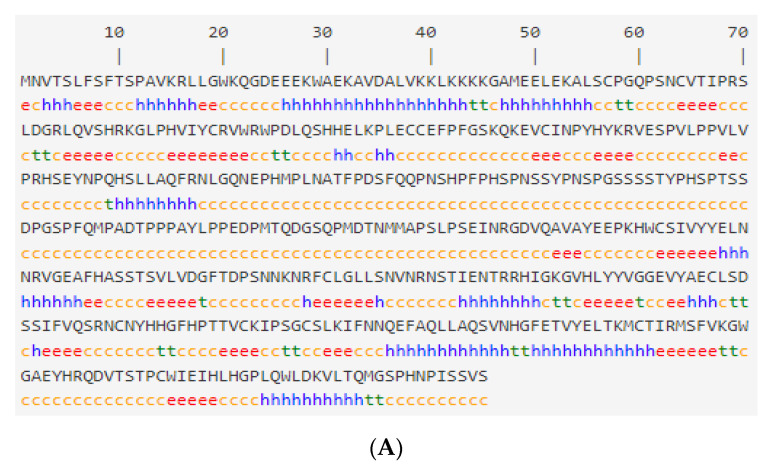
Protein secondary structure of SMAD1 (**A**), SMAD2 (**B**) and SMAD3 (**C**) in Tibetan sheep. The α helix is in blue, the β turn is in green, the random coil is in yellow, and the extended strand is in red.

**Figure 4 genes-13-02307-f004:**
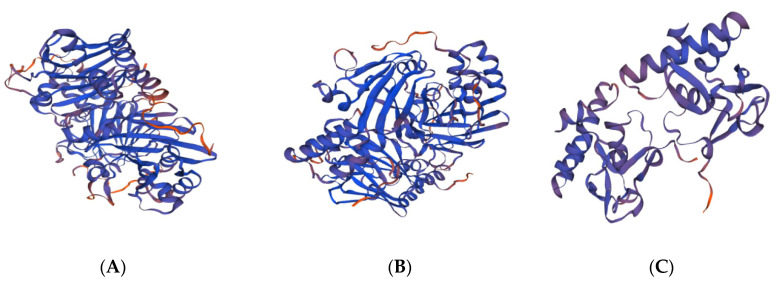
Protein tertiary structure of SMAD1 (**A**), SMAD2 (**B**) and SMAD3 (**C**) in Tibetan sheep.

**Figure 5 genes-13-02307-f005:**
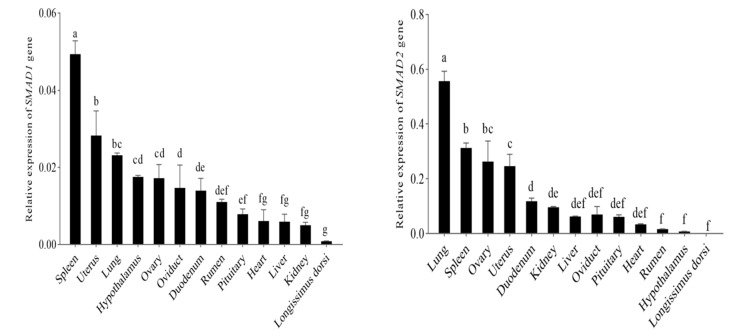
The mRNA expression of *SMAD1*, *SMAD2* and *SMAD3* genes in different tissues of Tibetan sheep. Bars with different letters indicate values with significant differences (n = 3) (*p* < 0.05).

**Figure 6 genes-13-02307-f006:**
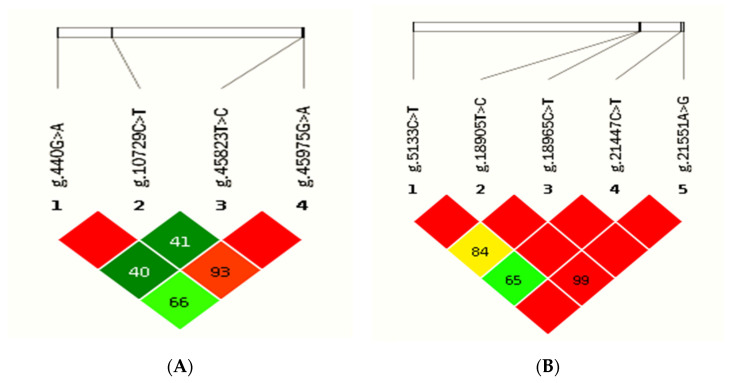
Linkage disequilibrium analysis of single nucleotide polymorphisms (SNPs) in *SMAD1* (**A**) and *SMAD3* (**B**) genes. The number in the blank indicates the D′ value (%).

**Table 1 genes-13-02307-t001:** The primer sequence information used for the cDNA cloning and expression analysis of *Smad1*, *Smad2* and *Smad3*.

Gene	GenBank ID	Primer Sequence (5′~3′)	Product Size (bp)	*T*m (°C)	Application
*SMAD1*	XM_027956253.2	TTTCAGGACCTCCGCACGAA CTGAATCCGACAGTTGGTCAC	1657	59	Cloning
*SMAD2*	XM_027960887.2	CCACCGCTTTTGGTAAGAACA ACCAATTCCACAAGGTGCTTT	1490	56	Cloning
*SMAD3*	XM_042252071.1	AACCAGCAAGTTCGCCGAGA GCCCTCTTCCCCAGATCCAC	1473	62	Cloning
*SMAD1*	XM_027961247.1	F:TCATCCCGGGAGGTGGCAGA R:GACCTCCTTCAGCCGCTGGT	75	64	qPCR
*SMAD2*	XM_027960887.1	F:AGGGTGGGGAGCAGAATACCG R:TTGTCCAACCACTGTAGAGGTCCA	93	63	qPCR
*SMAD3*	XM_027971520.1	F:CCCAGCCACCGTCTGCAAGAT R:CCAAGAGGGCGGCGAACTCC	77	65	qPCR
*GAPDH*	NM_001190390.1	F:GCGAGATCCTGCCAACATCAAGT R:CCCTTCAGGTGAGCCCCAGC	105	65	Reference gene

**Table 2 genes-13-02307-t002:** iMLDR genotyping primer.

Gene	Locus	PCR Primer Sequence (5′~3′)	LDR Primer Sequence (5′~3′)
*SMAD1*	g.45975G>A	F: 5′ACACAGATCTGCTCAATGCCTCTAC3′ R: 5′GAAGAGACTTCTTGGCTGGAAACAG3′	3′-end marker:TGAGCTGCCCGGGCCAGCTTTTTTTTTTTTTTTTTTTTTTT Allele 1:TGTTCGTGGGCCGGATTAGTTTCATGGAGGAACTGGAAAAGGACT Allele 2:TGTTCGTGGGCCGGATTAGTCATGGAGGAACTGGAAAAGGACC
	g.440G>A	F:5′GTGAGCCCATCTGGGTAAGGAC3′ R: 5′GCAGGTTAAATGACTCCCACTCTTG3′	3′-end marker:GAGATCCATCTGCACGGCCCTTTTTTTTTTTTTTTTTTT Allele 1:TGTTCGTGGGCCGGATTAGTTTCCAGCACSCCCTGCTGGCTT Allele 2:TGTTCGTGGGCCGGATTAGTCCAGCACSCCCTGCTGGTTC
	g.45823T>C	F: 5′ACACAGATCTGCTCAATGCCTCTAC3′ R: 5′GAAGAGACTTCTTGGCTGGAAACAG3′	3′-end marker:GGYTTCAGTTCRTGGTGGCTTTTTTTTTTTTTTTTTTTTTTTT Allele 1:TCTCTCGGGTCAATTCGTCCTTAAGGGAAACTCGCAGCATTCCTAC Allele 2:TCTCTCGGGTCAATTCGTCCAAGGGAAACTCGCAGCATTCCTAT
	g.10729C>T	F: 5′GTTGGAAGGATCGGTGAAACCAT3′ R: 5′CCAACTTTACGCCACAGTTTCAGT3′	3′-end marker:GGCTCCTCATARGCAACCGCTTTTTTTTTTTTTTTTTTTTTTTTTTTTTTTTTTTTTT Allele 1:TTCCGCGTTCGGACTGATATGAGCTCATAGTAGACAATAGAGCACCAGTGTGTC Allele 2:TTCCGCGTTCGGACTGATATTTGAGCTCATAGTAGACAATAGAGCACCAGTGTGTT
*SMAD2*	g.14946G>A	F: 5′CGTTGGAGAGTAAACCTAGGCAGAAC3′ R: 5′CAGACTTGCAGCCAGTTACTTACTCAG3′	3′-end marker:TCTACGGTGAGTGAGGGCTGTGTTTTTTTTTTTTTTTTTTTTTTTTTTTTTTTTTT Allele 1:TTCCGCGTTCGGACTGATATTTCAGAATTTGATGGATCTGTGAAGACA Allele 2:TTCCGCGTTCGGACTGATATCAGAATTTGATGGATCTGTGAAGACG
*SMAD3*	g.21447C>T	F: 5′AAGATCTTCAACAACCAGGAGTTC3′ R: 5′GGCCARCCTAGCTCATCTCTG3′	3′-end marker:GTCAACCAGGGCTTCGAGGCTTTTTTTTTTTTTTTTT Allele 1:TTCCGCGTTCGGACTGATATCCGCCCTCTTGGCTCAGACC Allele 2:TTCCGCGTTCGGACTGATATTTCCGCCCTCTTGGCTCAGACT
	g.5133C>T	F:5′ACTGTGATGTACACATCCTGTCATCTG 3′ R: 5′AGGGCTACAGCTAAAGGGGGTTC3′	3′-end marker:TGGGTAAGTAGYTCCTTATCTGTATGATTTGTTTTTTTTTTTTTTTTTTTTTTTTTTTTTTTTTTTTT Allele 1:TACGGTTATTCGGGCTCCTGCGATGTCCCCAGCACACAATATCC Allele 2:TACGGTTATTCGGGCTCCTGTTCGATGTCCCCAGCACACAATACCT
	g.18965C>T	F: 5′- CCTAGTTTAGCAGCCCAGCATCAC3′ R: 5′GGTACCTGGTGGAATCTTGCAGAC3′	3′-end marker:TTGGGAGACTGCACAAAGATGGTTTTTTTTTTTTTTTTTTTTTTTTT Allele 1:TGTTCGTGGGCCGGATTAGTCCAGCCATAGCGCTGGTTAGAG Allele 2:TGTTCGTGGGCCGGATTAGTTTCCAGCCATAGCGCTGGTTAGAA
	g.18905T>C	F: 5′CCTAGTTTAGCAGCCCAGCATCAC3′ R: 5′GGTACCTGGTGGAATCTTGCAGAC3′	3′-end marker:ATGTAGTACAGCCGCACRCCTTTTTTTTTTTTTTTT Allele 1:TACGGTTATTCGGGCTCCTGTTCTGCGAAGACCTCCCCTACG Allele 2:TACGGTTATTCGGGCTCCTGCTGCGAAGACCTCCCCTGCA
	g.21551A>G	F: 5′AAGATCTTCAACAACCAGGAGTTC3′ R: 5′GGCCARCCTAGCTCATCTCTG3′	3′-end marker:CCTGTCCTAGGGSCGCAGTTTTTTTTTTTTTT Allele 1:TACGGTTATTCGGGCTCCTGTACAGGTGGGTGCTGGGAYA Allele 2:TACGGTTATTCGGGCTCCTGTTTACAGGTGGGTGCTGGGAYG

Note: in primers, Y, R and S represent annexed bases, which are, respectively, C/T, G/A, and C/G.

**Table 3 genes-13-02307-t003:** Genotype and allele frequencies of SNPs loci of *SMAD1*, *SMAD2* and *SMAD3* genes in Tibetan sheep.

Gene	Locus	Exon	Mutation Type	Genotype	Genotype Frequency (No.)	Allele	Allele Frequency
*SMAD1*	g.45975G>A	2	Synonymous mutations	GG	0.73 (318)	G	0.85
				GA	0.23 (101)	A	0.15
				AA	0.03 (14)		
	g.440G>A	7	Synonymous mutations	GG	0.92 (397)	G	0.96
				GA	0.08 (36)	A	0.04
				AA	0.00 (0)		
	g.45823T>C	2	Synonymous mutations	TT	0.07 (29)	T	0.24
				TC	0.35 (149)	C	0.76
				CC	0.58 (245)		
	g.10729C>T	5	Synonymous mutations	CC	0.96 (417)	C	0.98
				CT	0.04 (16)	T	0.02
				TT	0.00 (0)		
*SMAD2*	g.14946G>A	8	Synonymous mutations	GG	0.47 (205)	G	0.70
				GA	0.46 (193)	A	0.30
				AA	0.08 (35)		
*SMAD3*	g.21447C>T	7	Synonymous mutations	CC	0.65 (283)	C	0.81
				CT	0.31 (135)	T	0.19
				TT	0.03 (15)		
	g.5133C>T	4	Synonymous mutations	CC	0.42 (184)	C	0.64
				CT	0.45 (200)	T	0.36
				TT	0.13 (59)		
	g.18965C>T	6	Synonymous mutations	CC	0.73 (315)	C	0.85
				CT	0.25 (108)	T	0.15
				TT	0.02 (10)		
	g.18905T>C	6	Missense mutation	TT	0.12 (50)	T	0.33
				TC	0.42 (184)	C	0.67
				CC	0.46 (199)		
	g.21551A>G	7	Synonymous mutations	AA	0.12 (50)	A	0.33
				AG	0.43 (186)	G	0.67
				GG	0.45 (197)		

**Table 4 genes-13-02307-t004:** Population genetic analysis of SNPs loci of *SMAD1*, *SMAD2* and *SMAD3* genes in Tibetan sheep.

Gene	Locus	Homozygosity (*Ho*)	Heterozygosity (*He*)	Effective Allele Numbers (*Ne*)	Polymorphic Information Content (*PIC*)	χ^2^ Test
*SMAD1*	g.45975G>A	0.75	0.25	1.34	0.22	0.23
	g.440G>A	0.92	0.08	1.08	0.07	1.00
	g.45823T>C	0.63	0.37	1.58	0.30	0.79
	g.10729C>T	0.96	0.04	1.04	0.04	1.00
*SMAD2*	g.14946G>A	0.58	0.42	1.73	0.33	0.31
*SMAD3*	g.21447C>T	0.69	0.31	1.45	0.26	1.00
	g.5133C>T	0.54	0.46	1.85	0.35	0.40
	g.18965C>T	0.75	0.25	1.34	0.22	0.85
	g.18905T>C	0.56	0.44	1.79	0.34	0.51
	g.21551A>G	0.56	0.44	1.79	0.34	0.51

*df* = 1, χ^2^_0.05_ = 3.841; *df* = 2, χ^2^_0.05_ = 5.991.

**Table 5 genes-13-02307-t005:** Association analysis between SNPs polymorphism of *SMAD1*, *SAMD2* and *SMAD3* genes and litter size in Tibetan sheep.

Gene	Locus	Genotype	No. of Individuals	Litter Size
*SMAD1*	g.45975G>A	GG	318	1.07 ± 0.25
		GA	101	1.06 ± 0.24
		AA	14	1.14 ± 0.36
	g.440G>A	GG	397	1.07 ± 0.25
		GA	36	1.11 ± 0.32
		AA	0	0.00 ± 0.00
	g.45823T>C	TT	29	1.07 ± 0.26
		TC	159	1.05 ± 0.22
		CC	245	1.08 ± 0.27
	g.10729C>T	CC	417	1.07 ± 0.26 ^a^
		CT	16	1.00 ± 0.00 ^b^
		TT	0	0.00 ± 0.00
*SMAD2*	g.14946G>A	GG	205	1.07 ± 0.25
		GA	193	1.08 ± 0.28
		AA	35	1.00 ± 0.00
*SMAD3*	g.21447C>T	CC	283	1.06 ± 0.24 ^b^
		CT	135	1.06 ± 0.24 ^b^
		TT	15	1.27 ± 0.46 ^a^
	g.5133C>T	CC	184	1.09 ± 0.28
		CT	190	1.04 ± 0.20
		TT	59	1.10 ± 0.30
	g.18965C>T	CC	315	1.07 ± 0.26
		CT	108	1.07 ± 0.26
		TT	10	1.00 ± 0.00
	g.18905T>C	TT	50	1.12 ± 0.33
		TC	184	1.05 ± 0.23
		CC	199	1.07 ± 0.26
	g.21551A>G	AA	50	1.12 ± 0.33
		AG	186	1.05 ± 0.23
		GG	197	1.07 ± 0.26

At the same site, different lowercase letters of shoulder tags in the same column data indicate significant differences (*p* < 0.05); the same letters or no letters on the shoulder tags indicate that the differences are not significant (*p* > 0.05).

**Table 6 genes-13-02307-t006:** Haplotypes of *SMAD1* and *SMAD3* genes SNPs loci.

Gene	SNPS Loci	Haplotype	Haplotype Frequency	OR	Std_Error	*p*-Value
*SMAD1*	g.440G>A, g.10729C>T	AC	0.04	1.855	0.5691616	0.2776538
		GC	0.94	-	-	-
		GT	0.02	0.0000003	0.9890451	0.9879198
	g.45823T>C, g.45975G>A	CA	0.16	1.193868	0.351637	0.6243151
		CG	0.71	-	-	-
		TG	0.26	0.8058256	0.3359107	0.5204223
*SMAD3*	g.5133C>T, g.18905T>C	CC	0.36	0.9627407	0.3181858	0.9050088
		CT	0.36	1.492368	0.3058867	0.1905804
		TC	0.61	-	-	-
	g.18905T>C, g.18965C>T, g.21447C>T, g.21551A>G	CCCG	0.59	-	-	-
		CCTG	0.23	1.932429	0.3571337	0.06509204
		CTCG	0.20	1.500328	0.4424184	0.3591594
		TCCA	0.37	1.599593	0.3127825	0.1331385

## Data Availability

Not applicable.

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
