# Peer review of "Expression and Polymorphisms of SMAD1, SMAD2 and SMAD3 Genes and Their Association with Litter Size in Tibetan Sheep (Ovis aries)"

_genes, 2022, doi:10.3390/genes13122307_

Round 1

Reviewer 1 Report

The present research studies "SMAD1, SMAD2 and SMAD3 Genes and Its Associations with Litter Size trait in Tibetan Sheep". This research provides interesting information. However, some changes need to be made.

Introduction

Line 45.- Is this sentence correct? "the lambing rate is only 105%".

Lin 87-89.- I suggest adjusting the objective “The SNPs of SMAD1, SMAD2 and SMAD3 genes were genotyped by iMLDR technology to analyze the association between the polymorphisms of genes and the litter size of Tibetan sheep.”

Material and methods

Line 91.- What animal welfare guidelines and norms did you use, did you have an ethics committee that supported the research?

Line 93.- they mention "aged" 6 months, why did they select animals with that age? How did you determine that they were not prepubertal animals. Also, it would be convenient to mention the average weights and body condition.

Line 194.- it would be convenient to mention if the data came from a population with normal distribution and if they had homoscedastic variances, what tests were performed?

Results

Line 262.- figures 1 and 2, the literals are very small and difficult to understand. I recommend enlarging it.

Line 291.- figure 3 literals are very small and difficult to understand. I recommend enlarging it.

Discussion

General comments:

Expand the discussion of the results of "Protein structure prediction of SMAD1, SMAD2 and SMAD3".

 Learn more about "ORFs" and their relationship with SMAD1, SMAD2 and SMAD3 genes.

Learn more about mRNA expression of the SMAD1, SMAD2 and SMAD3.

Conclusions

I recommend restructuring it, i.e., making it more concise and avoiding mentioning results that have already been mentioned previously. In addition to mentioning the possible implications and importance of this research.

Reviewer 2 Report

This is an interesting manuscript focused to explore the expression pattern and polymorphisms of SMAD1, SMAD2 and SMAD3 genes and its associations with litter size in Tibetan sheep (Ovis aries). The methodology is described with sufficient detail. Results are clear and concise, as well as discussion.

I only suggest considering next minor comments to improve the manuscript:

1)             Abstract: The extension of this section is too long (i.e., 340 words). According to guidelines described in “Instructions for authors”, the abstract should be a total of about 200 words maximum.

2)             Conclusions: I suggest to present this section as final conclusive statements (i.e., 5 to 6 sentences), instead of a summary of results.

Minor grammar comments:

-       Line 23: Reduce the letter size of the word “The”.

-       Line 45: Please check if the value of 105% of lambing rate is correct.

-       Line 68: Replace the comma by a period and remove the word “and”.

-       Line 71: Replace the semicolon by a period and use “By” instead of “by”.

-       Line 83: Replace “this” by “the current”.

-       Line 87: Replace semicolon by a period.

-       Line 119: Insert a comma between SMAD1 and SMAD2.

-       Line 121: Insert a comma between SMAD1 and SMAD2.

-       Line 136: Insert a comma between SMAD1 and SMAD2.

-       Lines 141-143: Something is missing in this sentence.

-       Line 143: Type a space between the sentences.

-       Line 155: Remove the second comma and replace by “and”.

-       Line 181: Remove the “a” and replace “was” by “were”.

-       Line 185: Insert “and” after the second comma.

-       Line 197: Replace “genotypic” by “Genotypic”.

-       Line 200: Type a space before the square bracket. Please apply this correction on the entire manuscript.

-       Line 201: Type a space before the round bracket. Please apply this correction on the entire manuscript.

-       Line 209: Reduce the letter size of the reference number.

-       Line 244: Type a space before and after the round bracket.

-       Line 252: Replace “close” by “closer”.

-       Lines 305, 311, 314 and 317: Remove the word “that”.

-       Line 306: Insert “was” after “expression”.

-       Line 640: Remove bold style of the reference number “45”.
